# Veratryl Alcohol Attenuates the Virulence and Pathogenicity of *Pseudomonas aeruginosa* Mainly via Targeting *las* Quorum-Sensing System

**DOI:** 10.3390/microorganisms12050985

**Published:** 2024-05-14

**Authors:** Songzhe Fu, Wenxu Song, Xiaofeng Han, Lin Chen, Lixin Shen

**Affiliations:** Key Laboratory of Resource Biology and Biotechnology in Western China, Ministry of Education, College of Life Science, Northwest University, Xi’an 710069, China; fusongzhe@126.com (S.F.); songwenxu@stumail.nwu.edu.cn (W.S.); hanxiaofeng2301@163.com (X.H.); chenlin@nwu.edu.cn (L.C.)

**Keywords:** *Pseudomonas aeruginosa*, quorum sensing, veratryl alcohol, virulence factors, pathogenicity

## Abstract

*Pseudomonas aeruginosa* is an opportunistic pathogen that usually causes chronic infections and even death in patients. The treatment of *P. aeruginosa* infection has become more challenging due to the prevalence of antibiotic resistance and the slow pace of new antibiotic development. Therefore, it is essential to explore non-antibiotic methods. A new strategy involves screening for drugs that target the quorum-sensing (QS) system. The QS system regulates the infection and drug resistance in *P. aeruginosa*. In this study, veratryl alcohol (VA) was found as an effective QS inhibitor (QSI). It effectively suppressed the expression of QS-related genes and the subsequent production of virulence factors under the control of QS including elastase, protease, pyocyanin and rhamnolipid at sub-inhibitory concentrations. In addition, motility activity and biofilm formation, which were correlated with the infection of *P. aeruginosa*, were also suppressed by VA. In vivo experiments demonstrated that VA could weaken the pathogenicity of *P. aeruginosa* in Chinese cabbage, *Drosophila melanogaster*, and *Caenorhabditis elegans* infection models. Molecular docking, combined with QS quintuple mutant infection analysis, identified that the mechanism of VA could target the LasR protein of the *las* system mainly. Moreover, VA increased the susceptibility of *P. aeruginosa* to conventional antibiotics of tobramycin, kanamycin and gentamicin. The results firstly demonstrate that VA is a promising QSI to treat infections caused by *P. aeruginosa*.

## 1. Introduction

Antibiotic resistance of bacteria has increased worldwide dramatically, leading to a declining cure rate and rising mortality in patients [1]. About 1.27 million people in the world have died directly from infection caused by bacteria with antibiotic resistance in 2019, and the number is expected to rise to 10 million by 2050 [2,3]. Exploring new anti-infective drugs or therapeutics is urgently needed to address this global crisis. *Pseudomonas aeruginosa* is a Gram-negative opportunistic pathogen that ubiquitously exists in various environments [4]. It is a major cause of hospital-acquired infections and usually leads to chronic and acute infections in immunocompromised patients, particularly those with burns [5] and cystic fibrosis [6], intubated patients and those with AIDs [7,8]. Treatment of infections caused by *P. aeruginosa* is very challenging and can even be life-threatening due to its inherent and acquired resistance to a broad spectrum of antibiotics including carbapenems [9,10]. The mortality rate caused by *P. aeruginosa* in immunocompromised patients is as high as 44.6%. The World Health Organization (WHO) has identified it as one of the priority pathogens that urgently needs to develop new anti-infective drugs [11]. One attractive anti-infection approach is inhibiting the infection without imposing any selective pressure on bacteria.

Quorum sensing (QS) is a cell–cell communication system used by bacteria to regulate collective behaviors [12]. The QS system, consisting of autoinducer encoding genes and transcriptional regulator protein production genes, produces and releases small molecules called auto-inducers (AIs) in a cell density-depending pattern [13]. When the amounts of AIs reach a certain threshold, they subsequently modify their gene expression to adjust their behaviors. This includes altering the production of virulence factors, enhancing motility and promoting biofilm formation [14].

*P. aeruginosa* contains three interconnected QS systems, the quinolone-based *pqs* system and acyl homoserine lactone (AHL)-based *rhl* and *las* systems [15,16]. Signal molecules of N-3-oxo-dodecanoyl-l-homoserine lactone (3-oxo-C12-HSL) and N-butanoyl-L-homoserine lactone (C4-HSL) separately derived from LasI and RhlI bind to the corresponding receptors of LasR and RhlR to regulate the downstream gene expression, like *lasB*, *rhlA* and *pqsA* [17]. The *pqs* system consists of the *pqs* operon *pqs*-*ABCDE* and the transcriptional regulator PqsR. The complex formed by the signal molecule of 2-heptyl-3-hydroxy-4-quinolone (PQS) and the regulator of PqsR regulates the gene expression of *phzABCDEFG*, *pqsA*, *rhlAB* and others [18]. 

The QS-regulated genes encode an array of virulence factors that directly contribute to the colonization, dissemination and outcome of events in individuals infected with *P. aeruginosa*. The *rhl*-regulated *rhlAB* directly encodes rhamnolipid synthesis, which is involved in biofilm formation in *P. aeruginosa* and alters the hydrophobicity of the host cells [19,20]. *las* controls *lasA* and *lasB*, which encode protease and elastase to degrade host proteins and enhance the colonization ability [21]. *pqs* is also involved in a series of virulence production and bacterial infection in *P. aeruginosa*. Biofilm formation is associated with chronic infections and drug resistance. Virulence factors and motility ability are mainly responsible for acute infections [22].

Interference with the QS system to reduce the virulence production and then attenuate or eliminate bacterial infections has become a novel strategy in the era of growing antibiotic resistance of bacteria. Substances that inhibit the QS system are called quorum-sensing inhibitors (QSIs) [23]. Quenching the QS system via QSIs could reduce the production of virulence factors without affecting the bacteria growth and improve the sensitivity to antibiotics [24]. This method leads to a reduction in bacterial toxicity and a decreased tendency for antibiotic resistance. 

White-rot fungi, belonging to the Basidiomycetes, always produce ligninolytic enzymes to degrade organic pollutants and different bioactive secondary metabolites with antimicrobial or anticancer properties [25]. Veratryl alcohol (VA) is one of the secondary metabolites of white-rot fungi, which could be used as a substrate for assessing the activity of lignin peroxidase [26]. In this study, VA was identified as a QSI against *P. aeruginosa*. At sub-inhibitory concentrations, VA has been shown to effectively suppress QS and its associated virulence traits. This includes a reduction in the production of virulence factors such as elastase, pyocyanin and rhamnolipid, as well as a decrease in motility and biofilm formation, all in a dose-dependent manner. Moreover, VA has demonstrated the ability to mitigate the pathogenic effects of *P. aeruginosa* in Chinese cabbage, *Drosophila melanogaster* and *Caenorhabditis elegans* compared with the control group with no VA treatment. VA might interact with the *las* system, mainly LasR, to exhibit anti-QS potential against *P. aeruginosa* based on molecular docking, the analysis of virulence gene expression in *lux*-based QS quintuple mutants and the QS mutant infection analysis. The results indicated that VA could inhibit the QS system of *P. aeruginosa* to attenuate the QS-regulated virulence and pathogenicity by targeting the *las* system, primarily through LasR. VA could be utilized as a potential anti-QS compound for treating *P. aeruginosa* infections.

## 2. Materials and Methods

### 2.1. Bacterial Strains, Chemicals and Culture Conditions

The plasmids and strains utilized in this study were detailed in Table 1, which included both *P. aeruginosa* and *Escherichia coli* OP50. These bacteria were routinely cultivated at a temperature of 37 °C on Luria-Bertani (LB) agar plates, or, alternatively, in LB broth with continuous shaking at a rate of 200 revolutions per minute (rpm). When necessary, mutant strains were supplemented with Trimethoprim (Tmp, at a concentration of 300 µg/mL) and Kanamycin (Kan, at a concentration of 50 µg/mL). All antibiotics and the signal molecules 3-oxo-C12-homoserine lactone (odDHL), N-butanoyl-L-homoserine lactone (BHL) and Pseudomonas quinolone signal (PQS) were obtained from Sigma-Aldrich (Sigma, St. Louis, MA, USA). Additionally, veratryl alcohol (VA) was sourced from Shanghai Yuanye Bio-Technology Co., Ltd. (Shanghai, China) and was subsequently diluted in double-distilled water for experimental purpose.

### 2.2. Minimum Inhibitory Concentration (MIC) Determination

Serial dilutions were performed to determine the MIC of VA by measuring the growth of *P. aeruginosa* at OD_600_ in clear 96-well plates (Costar, New York, NY, USA) using a Synergy H1microplate reader (BioTeck, Winooski, VT, USA). MICs were defined as the lowest concentration of VA inhibiting visible growth after 24 h of incubation at 37 °C.

### 2.3. Evaluation of the Transcriptional Expression of Genes Associated with QS

The expression levels of the QS-associated genes, including *lasI*, *lasR*, *rhlI*, *rhlR*, *lasB*, *pqsA*, etc., were monitored using the *lux*CDABE–reporter system as described in a previous study [31]. In brief, the plasmid of pMS-402, which contains a promoterless *luxCDABE* reporter gene cluster, was employed to generate promoter–*lux*CDABE reporter fusions for the QS-associated genes. The promoter regions of the genes were amplified and ligated into pMS-402, and then these reporter fusions were electroporated into *P. aeruginosa* respectively to obtain the *lux*CDABE–reporter reporter strains. The expression of these genes could be evaluated via the light production from the promoterless *lux*CDABE operon, which is in the downstream of the gene promoter. Overnight culture of these reporter strains was diluted to an OD_600_ of 0.2 and then cultured for 3 h. An aliquot of 5 µL of the cultures was inoculated separately into the wells containing 95 µg/mL of LB broth with varying concentrations of VA in a black clear-bottom 96-well plate. Each concentration should be in three parallels, and LB incubated with ddH_2_O was used as a control. A total of 50 µL of filter-sterilized paraffin was added to each well to prevent evaporation. The luminescence and growth of the report strains were monitored every 30 min for 24 h using a Synergy H1microplate reader.

### 2.4. The Production of the QS-Associated Virulence Factors

#### 2.4.1. Elastase

The production of elastase from *P. aeruginosa* was quantified using the Elastin-Congo red assay according to the previous report [32]. Briefly, 500 µL overnight culture of *P. aeruginosa* was inoculated into 5 mL LB broths containing different concentrations of VA (0, 1/4MIC, 1/2MIC) and incubated at 37 °C for 24 h with shaking. The cultures were centrifuged at 10,000 rpm for 10 min and then filtered via 0.22 µm membrane filters to obtain supernatants. 

A total of 200 µL of the supernatants was added to 800 µL of the Elastin-Congo red buffer (5 mg/mL elastin Congo red, 100 mM Tris-HCl, 1 mM of CaCl_2_, pH = 7.5) and then incubated with shaking at 37 °C for 6 h. Insoluble ERC was pelleted by centrifugation. Elastase activity was measured by detecting OD_495_ of the supernatants.

#### 2.4.2. Protease

Protease activity was determined by using an azocasein assay [27]. A total of 200 µL of filtered supernatants derived from the above groups was added to 800 µL phosphate buffer (pH = 7.8) containing 2% non-fat milk powder. The mixtures were incubated at 37 °C for 30 min to determine the absorbance of the solution at 440 nm.

#### 2.4.3. Pyocyanin Assay

Pyocyanin was extracted from the culture supernatants using the procedure detailed by Essar et al. [33]. *P. aeruginosa* was cultured in LB broth containing different concentrations of VA (0, 1/4MIC, 1/2MIC) with shaking for 17 h at 37 °C. A total of 5 mL of the cultured supernatants was mixed with 3 mL of chloroform. The chloroform phase was transferred into 1 mL of 0.2 N HCl for mixing thoroughly. After removing the HCl layer, the optical density of the pink phase solution was measured at 520 nm.

#### 2.4.4. Rhamnolipid

The production of rhamnolipid was determined by mineral salt cetyl trimethyl ammonium bromide (CTAB)-methylene blue agar test [34]. Briefly, 10 µL of overnight culture of *P. aeruginosa* was spotted on the center of the CTAB-MB plates (0.2 mg/mL CTAB, 5 µg/mL methylene blue) containing different concentrations of VA (0, 1/4MIC, 1/2MIC). The plates were incubated at 37 °C for 24 h. The production of rhamnolipid was assessed according to the diameter of the halo around the colony. 

### 2.5. Determination of Motility

Bacterial motility of swimming, swarming and twitting were assayed as previously described [35]. 

Briefly, 2 µL of overnight culture of *P. aeruginosa* was spotted at the center of swimming (nutrient broth 1%, glucose 0.5%, and agar 0.3%) and swarming (0.3% agar, 1% tryptone, 0.5% yeast extract powder and 0.5% sodium chloride) agar plates containing different concentrations of VA (0, 1/4MIC, 1/2MIC). The plates were incubated at 37 °C for 24 h, and the diameter of the motility zones was measured. Thin LB agar plates containing different concentrations of VA (0, 1/4MIC, 1/2MIC) were used for the twitching motility assay. A total of 2 µL of overnight cultured of *P. aeruginosa* was stabbed into the bottom of LB agar and then incubated at 37 °C for 24 h. The twitching bacterial cells were stained with crystal violet and washed with PBS to determine diameters.

### 2.6. Qualification of Biofilm Formation

Biofilm formation was determined in microtiter plates as described previously [36]. A total of 300 µL of overnight culture of *P. aeruginosa* was inoculated into each well containing 3 mL of LB with different concentrations of VA (0, 1/4MIC, 1/2MIC) in a 12-well plate (Corning/Costar, Corning, NY, USA). The plate was incubated at 37 °C for 24 h to facilitate biofilm formation. The plate was washed three times by PBS to remove any loosely attached bacterial cells. The biofilm was stained by 1% crystal violet for 20 min at room temperature. Excess crystal violet was washed by ddH_2_O. Biofilms stained with crystal violet were washed with 1 mL of 95% ethanol, and the amount of biofilm biomass was measured at OD_595_. 

### 2.7. In Vivo Pathogenicity Assay

Chinese cabbage, *Drosophila melanogaster* and *Caenorhabditis elegans* (*C. elegans*) models were used to determine the effect of VA on pathogenicity of *P. aeruginosa*.

#### 2.7.1. Chinese Cabbage Infection Assay

The Chinese cabbage infection assay was investigated as reported previously [37]. *P. aeruginosa* was cultured in LB broth containing different concentrations of VA (0, 1/4MIC, 1/2MIC) at 37 °C for 24 h. The cell pellets were collected by centrifugation and then resuspended in 10 mM MgSO_4_ to 2 × 10^7^ CFU mL^−1^. A total of 10 µL of the suspensions was injected into the midribs of the cabbage stems, which have been sterilized by 0.1% H_2_O_2_. The cabbage was then placed into the petri dishes containing a Whatman filter soaked with 10 mM MgSO_4_. The plates were kept at 30 °C for 5 d and then the rotten area of cabbage was monitored.

#### 2.7.2. *Drosophila melanogaster* Infection Assessment

Fruit fly (*D. melanogaster* Canton S) infection assessment was performed according to the previous report [38]. Overnight cultures of *P. aeruginosa* from LB broth containing different concentrations of VA (0, 1/4MIC, 1/2MIC) were centrifugated at 8000 rpm for 5 min to pellet cells. The cells were then suspended to OD_600_ of 2.0 by 5% sucrose. A total of 100 µL of the resuspension was spotted onto the sterility filter disk, which fully covered the surface of the sucrose agar (5% sucrose, 1% agar) in a vial. In total, 30 male fruit flies (1–3 d old) which have been starved for 3 h were transferred into each vial. The vials were kept at 25 °C, and the number of survived fruit flies was counted every 24 h for 10 days.

#### 2.7.3. *Caenorhabditis elegans* Infection Analysis 

A total of 200 µL of overnight culture of *P. aeruginosa* was spread on NGM plates (0.3% NaCl, 0.25% tryptone, 1 mM MgSO_4_, 1 mM CaCl_2_, 5 µg/mL cholesterol, 100 µg/mL FUDR, and 2% agar) containing different concentrations of VA and incubated for 24 h at 37 °C to obtain a bacterial lawn using plates coated by *E. coli* OP50 as controls. Synchronized N2 hermaphrodite worms at L4 phase were seeded on the NGM plates. The survival rate was measured every 12 h for 5 days [39]. 

### 2.8. Molecular Docking Analysis

Molecular docking analysis was conducted in accordance with a previous study [40]. The structural information for VA was sourced from PubChem (https://pubchem.ncbi.nlm.nih.gov, accessed on 18 December 2023). Additionally, the three-dimensional (3D) structures of the LasR (PDB ID: 6D6A), LasI (PDB ID: 1RO5), PqsA (PDB ID: 5OE6) and PqsR (PDB ID: 4JVD) proteins were retrieved from the Protein Data Bank (https://www.rcsb.org, accessed on 18 December 2023). Utilizing AutoDock Vina, the 3D-crystal structure of VA was docked into the respective binding sites of these proteins. The outcomes of the docking simulations were visualized using the PyMOL Molecular Graphics System, version 2.3.4. Furthermore, the root mean square deviation (RMSD) values for each docking exercise, based on the VA ligand, were computed using the same PyMOL Molecular Graphics System, version 2.3.4.

### 2.9. Exploration of the Main Target of VA on QS System

The three quintuple mutants of *P. aeruginosa*, PAO1 (Δ*lasI*Δ*rhlI*Δ*pqsA*Δ*rhlR*Δ*pqsR*) designated as QM-1, PAO1 (Δ*lasI*Δ*rhlI*Δ*pqsA*Δ*lasR*Δ*pqsR*) designated as QM-2 and PAO1 (Δ*lasI*Δ*rhlI*Δ*pqsA*Δ*lasR*Δ*rhlR*) designated as QM-3, in which only one receptor coding gene of *lasR*, *rhlR* or *pqsR* was contained, respectively, were used to detect the inhibited target of VA on the QS system of *P. aeruginosa*. The reporter strains of QM-1 (pKD-*lasB*), QM-2 (pKD-*lasB*) and QM-3 (pKD-*pqsA*) were constructed as previously described. The gene expression of *las* and *rhl*-regulated *lasB* and *pqs*-regulated *pqsA* was determined in LB broth containing different concentrations of VA and 10 mM of the corresponding exogenous signaling molecules (odDHL, BHL, PQS). 

The *lasR* knockout mutant of PAO1 (Δ*lasR*) constructed by our lab previously was used to further evaluate the effect of VA on the pathogenicity of *P. aeruginosa* via Chinese cabbage infection analysis to verify the target of VA on the QS system of *P. aeruginosa*.

### 2.10. Synergistic Antibacterial Effects of VA with Antibiotics

An antibiotic–VA combination experiment was performed as previously described with some modifications [41]. The MICs of the antibiotics against *P. aeruginosa* were determined by CLSI [42]. Then, overnight cultures of *P. aeruginosa* were transferred to fresh LB broth and activated to the log phase, the bacteria were diluted and added to the 96-well plate with a final concentration of 5 × 10^5^ CFU/mL per well. The sub-MICs of antibiotics with different concentrations of VA were added to the well, the mixture was incubated at 37 °C for 24 h and the growth curve was determined by measuring the absorbance of OD_600_ every 1 h. 

### 2.11. Statistical Analysis

All experiments were conducted three times to ensure reproducibility, and the results are expressed as the mean ± standard deviation (SD). Statistical analysis was performed using GraphPad Prism version 5.0.1 to identify significant differences, with a threshold for statistical significance set at *p* < 0.05. The difference among growth was analyzed by one-way repeated-measures ANOVA, the difference on the survival rate by log rank and the others were analyzed by using *t* tests.

## 3. Results

### 3.1. VA Inhibited QS-Associated Gene Expression in P. aeruginosa

The QS system plays key roles in the virulence of *P. aeruginosa*. In order to evaluate the anti-virulence potential of VA, the impact of VA on the expression of QS and QS-regulated virulence genes was assayed firstly. To exclude the possible effect of inhibited bacterial growth by VA on gene expression and virulence factor production, the MIC of VA against *P. aeruginosa* needs to be determined. The susceptibility was determined by the broth microdilution method. As shown in Figure 1, the MIC of VA against *P. aeruginosa* was 2048 µg/mL. VA at 1/2MIC (1024 µg/mL) and 1/4MIC (512 µg/mL) did not affect the growth of *P. aeruginosa* compared to the control without VA. Therefore, 1/2 MIC and 1/4 MIC of VA were used for the subsequent study.

The effects of VA at 1/4 and 1/2 of the MIC on the expression of genes involved in QS systems, including *lasI*, *lasR*, *rhlI*, *rhlR*, *pqsA* and *pqsR*, were determined at the transcriptional level. The results indicated that the expression of all the genes was reduced in a dose-dependent manner. The expressions of two genes, *lasI* and *pqsR*, were inhibited before 16 h but not after that, while the expressions of *lasR*, *rhlR*, *rhlI* and *pqsA* were inhibited for the entire period of observation, as shown in Figure 2. Furthermore, the gene expression of *lasB* regulated by the *las* system, *rhlA* regulated by *rhl* and *PhzA1* regulated by the *pqs* system were also measured. Their expressions were decreased compared to the control, as shown in Figure 2. However, the growth of *P. aeruginosa* showed no difference in the presence of VA at 1/2 and 1/4 of the MIC compared to growth in the absence of VA. The results indicated that VA could inhibit the transcriptional expression of QS-associated genes at sub-inhibitory concentrations of 1/2 and 1/4 of the MIC without impacting the growth of *P. aeruginosa*.

### 3.2. VA Inhibited the Production of QS-Regulated Virulence Factors

*P. aeruginosa* produce and/or secrete an array of virulence factors, which result in the primary attachment, movement and subsequent infection of bacteria, and QS systems regulate them with a hierarchical pattern. The production of QS-related virulence factors, including *lasA*-coded protease, *lasB*-coded elastase, *rhlA*-coded rhamnolipid and *phzA*-operon-coded pyocyanin, were measured in LB medium containing 0, 1/2 and 1/4 of MIC of VA, as depicted in Figure 3. The production of elastase was significantly decreased by 52.0% and 50.4% after being treated with VA at 1/2MIC and 1/4MIC, respectively, compared to the control. Similarly, the productions of the other three virulence factors were also reduced by VA at sub-inhibitory concentrations in a dose-dependent manner compared with the control: 59.7% and 41.5% reduction for protease, 48% and 38.4% inhibition for pyocyanin, 49.5% and 31.7% decrease for rhamnolipid, respectively. The results are consistent with the inhibitory effect of VA on gene expression at the transcriptional level.

### 3.3. VA Inhibited Motility and Biofilm Formation of P. aeruginosa

Motility and biofilm formation are two typical important virulence-associated features controlled by the QS system in *P. aeruginosa*. Motility of *P. aeruginosa* is closely related to its attachment, invasion and chronic infection on the surface of host cells. Biofilm formation is a typical feature of chronic *P. aeruginosa* infection. The motility and biofilm formation capacities of *P. aeruginosa* were assessed in LB broth with and without VA at 1/2MIC and 1/4MIC. As shown in Figure 4, compared with the control, VA at used concentrations significantly reduced the motility capacity of *P. aeruginosa*. There was a 72.7% and 32.8% decrease in swarming (Figure 4A), 38.9% and 14.5% decrease in swimming (Figure 4B) and 68.5% and 38.9% decrease in twitching (Figure 4C). Accordingly, the biofilm formation at 24 h was inhibited by 64.3% and 35.4%, respectively in the presence of 1/2 and 1/4MIC VA compared with the control, as shown in Figure 5, and a similar result was observed for 48 h biofilm formation.

### 3.4. Attenuated Virulence and Pathogenicity of P. aeruginosa by Sub-Inhibitory VA

Chinese cabbage, *Drosophila melanogaster* and *C. elegans* infection models were used to investigate the effect of VA on the virulence and pathogenicity of *P. aeruginosa*. As shown in Figure 6A, in the Chinese cabbage infection model, the decayed area on the cabbage stem caused by *P. aeruginosa* was significantly decreased by 60.1% and 37.7% in the presence of 1/2 and 1/4MIC VA compared with the control after 6 days of inoculation. This suggests that VA could significantly decrease the virulence of *P. aeruginosa* in a dose-dependent manner. Also, the survival rate of *C. elegans* infected by *P. aeruginosa* was elevated by 55% and 40% in 1/2MIC and 1/4 MIC VA treatment groups compared with the group without VA treatment. This indicates that VA could attenuate the pathogenicity of *P. aeruginosa* in *C. elegans*. Similarly, VA could significantly decrease the mortality of *Drosophila melanogaster* caused by *P. aeruginosa*, demonstrating its constructive capability to inhibit the pathogenicity of *P. aeruginosa*. All the results indicate that VA could attenuate the virulence and pathogenicity of *P. aeruginosa* in vivo.

### 3.5. Docking Analysis for the Probable Targets of VA against the QS System of P. aeruginosa

To explore the possible targets of VA against the QS systems of *P. aeruginosa*, molecular docking analysis in silico was carried out to analyze the interaction of VA with the QS-associated proteins of LasI, LasR, PqsA and PqsR. As shown in Figure 7, VA could bind to LasR by H-bonds at residues of Tyr56 Tyr93 and Trp60 with a docking energy of −3.62 Kcal/moL and RMSD of 0.487. Similarly, VA might interact with LasI and PqsA by H-bonds at residues of Arg30 Val143 Thr144, Tyr211 Gly279 and Gly302, respectively, and their docking energies were −3.25 and −3.45 Kcal/moL with RMSD of 0.756 and 0.932, respectively. In addition, PqsR binds to VA with a docking energy of −3.25 Kcal/moL and RMSD of 0.756. However, due to the unavailability of 3D structure data for RhlR and RhlI in the PDB database, it was unable to perform docking studies of VA with these proteins. Therefore, VA could interact with all the proteins of the QS system, and the docking energy of LasR showed the highest binding energy with a convincing RMSD, suggesting that VA might probably inhibit the QS systems of *P. aeruginosa* mainly by targeting LasR.

### 3.6. VA Inhibits the QS System of P. aeruginosa Mainly by Targeting LasR 

To further verify the predominant target of VA’s impact on the QS systems of *P. aeruginosa*, three quintuple mutants of *P. aeruginosa*, PAO1 (Δ*lasI*Δ*rhlI*Δ*pqsA*Δ*rhlR*Δ*pqsR*) designated as QM-1, PAO1 (Δ*lasI*Δ*rhlI*Δ*pqsA*Δ*lasR*Δ*pqsR*) designated as QM-2 and PAO1 (Δ*lasI*Δ*rhlI*Δ*pqsA*Δ*lasR*Δ*rhlR*) designated as QM-3, in which only one QS receptor of LasR, RhlR or PqsR was contained, respectively, were constructed. These mutants were used to obtain the reporter strains of QM-1 (pKD-*lasB*), QM-2 (pKD-*lasB*) and QM-3 (pKD-*pqsA*) to detect the main inhibited target of VA on the QS system of *P. aeruginosa*. The gene expression of *lasB* regulated by *las* and *rhl* and *pqsA* regulated by *pqs* was determined in LB broth containing 1/2MIC or 1/4MIC of VA in the presence of 10 mM of the corresponding exogenous signaling molecules of odDHL, BHL and PQS, respectively. As shown in Figure 8, VA could significantly inhibit the expression of *lasB* in QM-1 (pKD-*lasB*) in a dose-dpendent pattern but not the expression of *lasB* in QM-2 (pKD-*lasB*) and that of *pqsA* in QM-3 (pKD-*pqsA*). The results suggested that VA inhibited the QS systems by interfering with the *las* system.

Then, the mutants of PAO1 (Δ*lasR*) were used to investigate the pathogenicity in Chinese cabbage treated with VA at 0, 1/2MIC and 1/4MIC. As shown in Figure 9, the pathogenicity of PAO1 (Δ*lasR*) was not obviously attenuated by VA, confirming that VA could inhibit QS of *P. aeruginosa* mainly by interacting with LasR.

### 3.7. Synergistic Bacteriostatic Effects of VA and Antibiotics 

The MICs of antibiotics Kan, Tob and Gm against *P. aeruginosa* were measured firstly as 60 µg/mL, 2 µg/mL and 30 µg/mL, respectively. The anti-bacterial effects of these antibiotics at 1/2MIC combined with VA were further determined. As shown in Figure 10A, treatment by Kan at 1/2MIC combined with VA (1/2MIC, 1/4MIC) significantly inhibited the growth of *P. aeruginosa* after 8 h compared with the group treated by Kan alone. Similar results were also obtained when combining 1/2 MIC Tob with VA and combining 1/2Tob with VA, shown in Figure 10B,C. The results indicated that VA improved the susceptibility of *P. aeruginosa* to antibiotics and suggested that the combination of VA with antibiotics could be an effective strategy for treating *P. aeruginosa* infection.

## 4. Discussion

*P. aeruginosa* is an opportunistic pathogen that often causes a variety of infections, occasionally resulting in death in patients. In recent years, the treatment of *P. aeruginosa* with traditional antibiotics has become increasingly difficult due to its rising antibiotics resistance [43]. Infectivity and antibiotic resistance of *P. aeruginosa* are associated with the secretion of virulence factors and biofilm formation, respectively, both of which are regulated by the QS system. Therefore, searching for compounds that target the QS system is a novel alternative option to control the infection of *P. aeruginosa.*

In this study, VA, a secondary metabolite of white-rot, has been proven to act a QSI by inhibiting the production of *P. aeruginosa* virulence factors of protease, elastease, rhamno-lipid and pyocyanin, as well as motility ability and biofilm formation, which are all regulated by QS, leading to the reduction in pathogenicity of *P. aeruginosa*. The other secondary metabolites of fungi have also been reported as QSIs. Actinomycin D, a metabolite from the endophyte *Streptomyces cyaneochromogenes* RC1, could inhibit the QS system in various ways and ultimately reduce the production of virulence factors and biofilm formation [44]. Terrein isolated from *Aspergillus terreus* exerts QSI activity by inhibiting both *las* and *rhl* systems as well as the synthesis of cellular c-di-GMP, demonstrating a noticeable protective effect in *C. elegans* and mice [45]. 

Traditional medicine is a type of natural product with a long history application in disease therapy. They usually show anti-inflammatory, anti-infective and other capacities. Some studies have been dedicated to discovering QSIs from traditional herbal medicines due to their high safety profile and low side effects. *Tanreqing* injection, *Yunnan Baiyao* water extract and the alcohol extract of burdock leaf, which have been used in clinic disease treatment, have also been reported to reduce QS system activity, which ultimately inhibits the production of virulence factors such as elastase and pyocyanin, reduces bacterial motility, and prevents biofilm formation [46,47,48]. Furthermore, some compounds derived from traditional herbs, such as luteolin from various herbs, paonol from *Moutan cortex*, perillaldehyde from *Perilla frutescens* and falcarindiol from *Notopterygium incisum*, have been reported as QSIs [49,50,51]. Our previous study demonstrated that wogonin and Sennoside A separately derived from Chinese herbs of *Agrimonia pilosa* and *R. palmatum L* exhibit QSI activity by targeting the *pqs* and *las* systems, respectively. 

However, when considering the clinical application of these non-pharmaceutical QSIs, it is important to take into account their cytotoxicity. The cytotoxic analysis revealed that VA exhibited toxicity towards pulmonary epithelial cells under experimental concentrations. Nevertheless, the nematode and fruit fly experiments indicate that VA at experimental concentrations did not affect the survival rate of them, similar to those of the blank control group (Figure 6B,C). The probable reason might be due to the multicellular organization and tissue structure of nematodes and fruit fly, which could eliminate cellular toxicity of VA, and the specific mechanism needs to be explored further. Further exploration is also required by using mouse models to determine the cytotoxicity of VA, QS inhibitory effect and the synergy effect of VA combining with antibiotics on *P. aeruginosa* infection. Similarly, for the chemically synthesized QSIs, which have not yet undergone clinical use, we also need to consider the cell toxicity before use.

The inhibitory mechanism of QSIs includes competitively blocking the binding of signal molecules to their corresponding receptor proteins, inhibiting the biosynthesis of QS signal molecules and degrading the signal molecules in cells [52]. VA might compete with signal molecular AHL for LasR protein to inhibit the QS system and attenuate the pathogenicity of *P. aeruginosa* based on molecular docking analysis as well as the quintuple mutants analysis. Furanone C-30 and a phosphate ester derivative of chrysin could act as structural analogues to bind to the active pocket of the LasR protein and subsequently destabilize the AHL-LasR dimer, leading to the dysregulation of the LasR protein [53,54]. Similarly, stigmatellin Y from *Bacillus subtilis* BR4 and vanillin from vanilla beans could competitively bind with PqsR to interfere with the *pqs* system [55,56]. Azithromycin at sub-MIC down-regulated the expression of N-acyl homoserine lactone (AHL) synthesis enzymes, leading to a decrease in AHL production and finally affecting the QS system of *P. aeruginosa* [57]. A series of enzymes have been reported to have AI-degrading activities. For instance, acylase AiiD cloned from *Ralstonia strain* XJ12B could degrade the AHL amide, resulting in a decrease in swarming ability and virulence production. Penicillin V acylases from Gram-negative bacteria and AHL acylase PfmA from *Pseudoalteromonas flavipulchra* JG1 could degrade the AHLs to inhibit the QS system in *P. aeruginosa* [58,59,60]. There are also QSIs that show a complex QS inhibition mechanism. Quercetin not only inhibits the production of AHL signal molecules but also competes with the signal molecule to bind with the receptors [61].

Despite the multiple action mechanisms of QSIs, bacteria could make adjustments to this adverse survival situation, leading to the invalidation of QSIs and becoming resistant to them. However, the development of this type of resistance is much slower compared to the resistance caused by antibiotics. Therefore, QSIs remain a potential drug to combat bacterial infections.

## 5. Conclusions

This study revealed that VA at a concentration ≤ 1024 µg/mL could function as a QSI to reduce the virulence factor production and the pathogenicity of *P. aeruginosa*. VA might mainly target the *las* system, probably inhibiting the binding of signal molecule AHL to LasR, to block the QS system. And VA could improve the susceptibility of *P. aeruginosa* to Kan, Tob and Gm. VA only or VA combining with these antibiotics could be an effective strategy for treating *P. aeruginosa* infection. However, the cytotoxicity of VA needs further investigating.

## Figures and Tables

**Figure 1 microorganisms-12-00985-f001:**
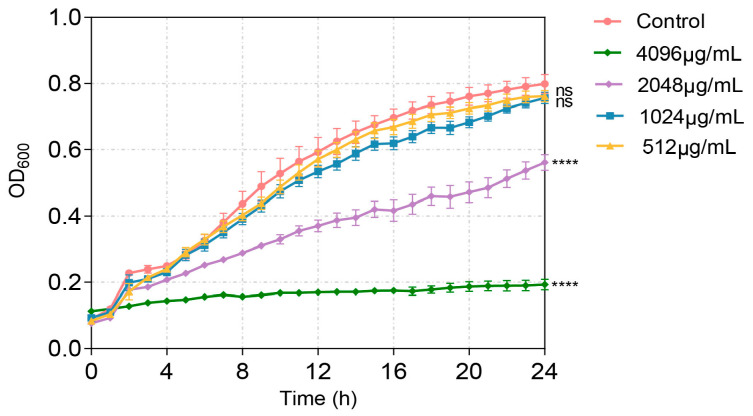
Effect of VA on the growth of *P. aeruginosa*. The data were expressed as mean ± SD values with three independent experiments. ns *p* > 0.05; **** *p* < 0.0001.

**Figure 2 microorganisms-12-00985-f002:**
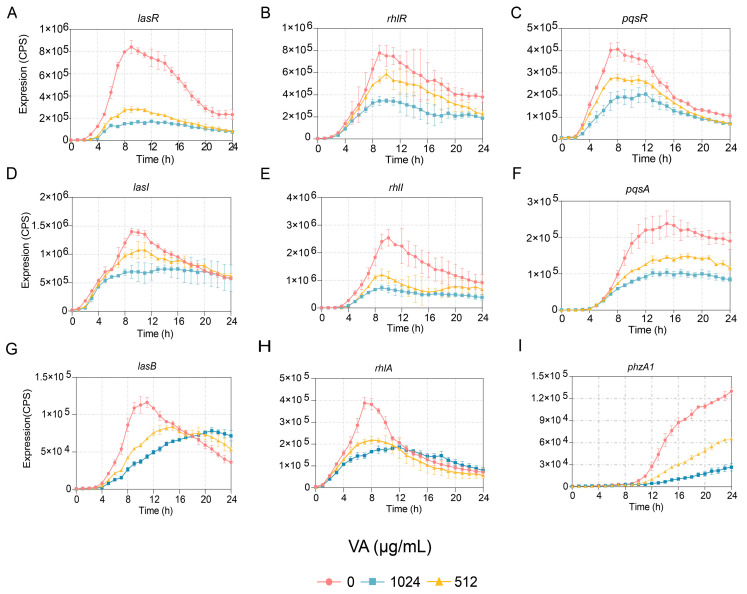
Effects of VA on the expression of QS-regulated genes in *P. aeruginosa*. (**A**) *lasR*; (**B**) *rhlR*; (**C**) *pqsR*; (**D**) *lasI*; (**E**) *rhlI*; (**F**) *pqsA*; (**G**) *lasB*; (**H**) *rhlA*; (**I**) *phzA1*.

**Figure 3 microorganisms-12-00985-f003:**
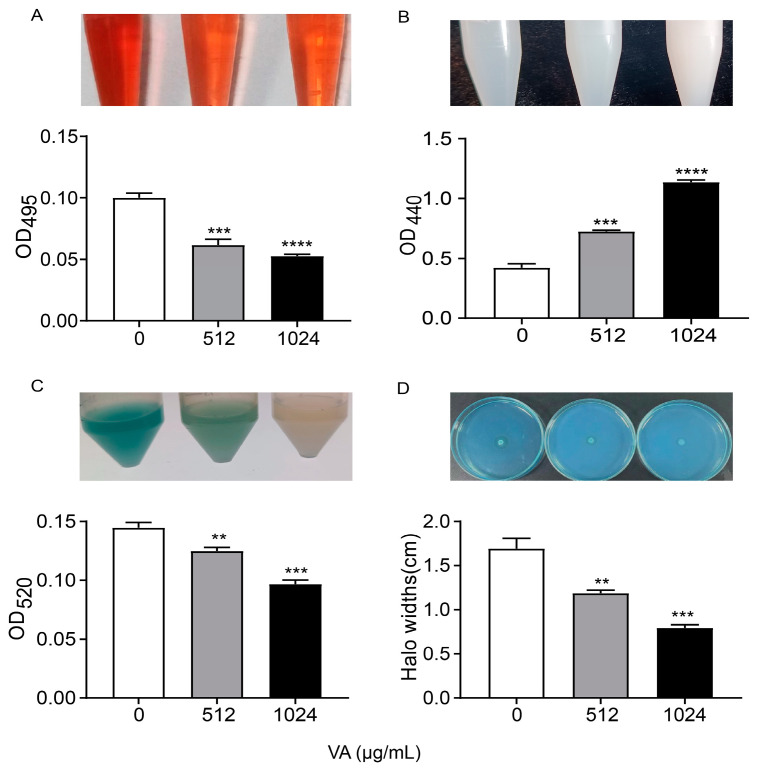
Effects of VA on virulence factors production in *P. aeruginosa*. (**A**) Elastase; (**B**) protease; (**C**) pyocyanin; (**D**) rhamnolipid. The concentration of VA was 512 µg/mL, 1024 µg/mL, respectively. ** *p* < 0.01; *** *p* < 0.001; ***** p* < 0.0001.

**Figure 4 microorganisms-12-00985-f004:**
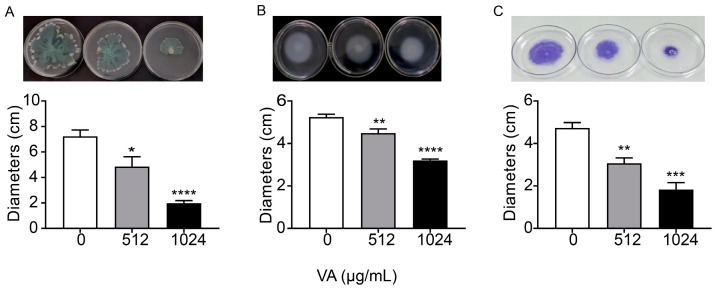
Effects of VA on the motilities in *P. aeruginosa*. (**A**) Swarming; (**B**) swimming; (**C**) twitching. The concentration of VA was 512 µg/mL, 1024 µg/mL, respectively. * *p* < 0.05; ** *p* < 0.01; *** *p* < 0.001; **** *p* < 0.0001.

**Figure 5 microorganisms-12-00985-f005:**
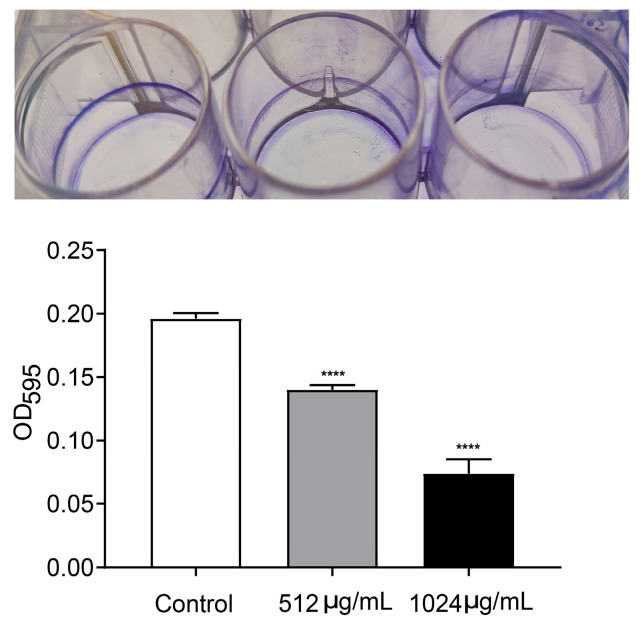
Effect of VA on biofilm formation in *P. aeruginosa*. The concentration of VA was 512 µg/mL, 1024 µg/mL, respectively. **** *p* < 0.0001.

**Figure 6 microorganisms-12-00985-f006:**
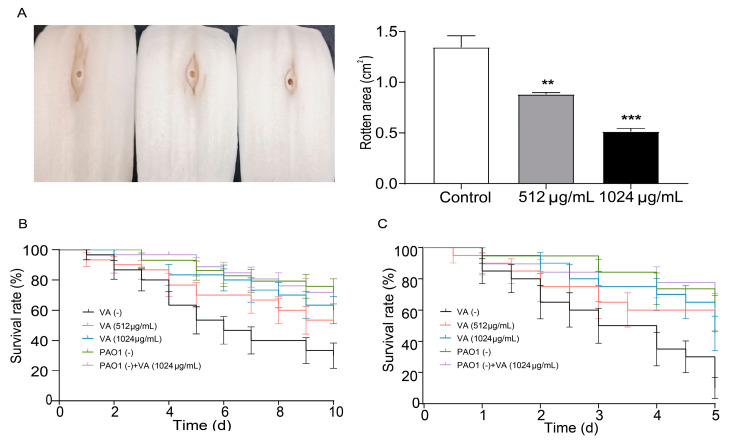
Effects of VA on the pathogenicity in *P. aeruginosa*. (**A**) Chinese cabbage; (**B**) *Drosophila melanogaster*; (**C**) *C. elegans*. The concentration of VA was 512 µg/mL, 1024 µg/mL, respectively. VA (-): without VA, PAO (-): without PAO1. ** *p* < 0.01; *** *p* < 0.001.

**Figure 7 microorganisms-12-00985-f007:**
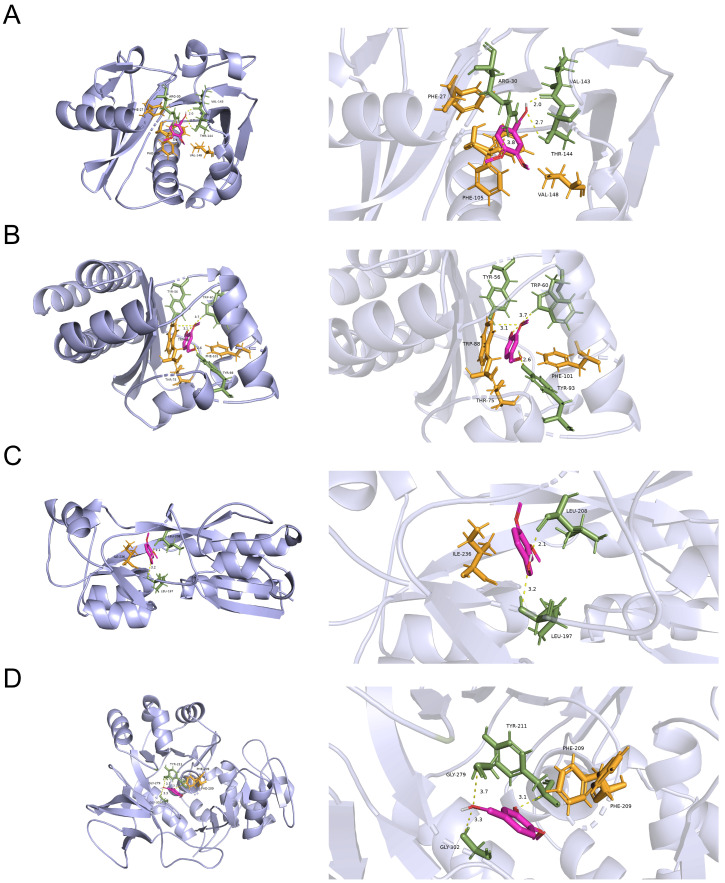
The molecular docking results of VA with (**A**) LasI protein, (**B**) LasR protein, (**C**) PqsR protein and (**D**) PqsA protein.

**Figure 8 microorganisms-12-00985-f008:**
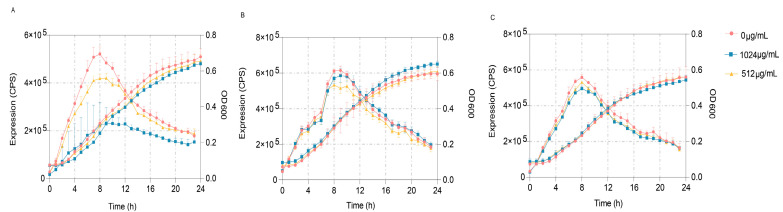
Effects of VA on the expression of *lasB* and *pqsA* in the QS quintuple mutants. (**A**) The expression of *lasB* in QM-1 (pKD-*lasB*); (**B**) the expression of *lasB* in QM-2 (pKD-l*asB*); (**C**) the expression of *pqsA* in QM-3 (pKD-*pqsA*).

**Figure 9 microorganisms-12-00985-f009:**
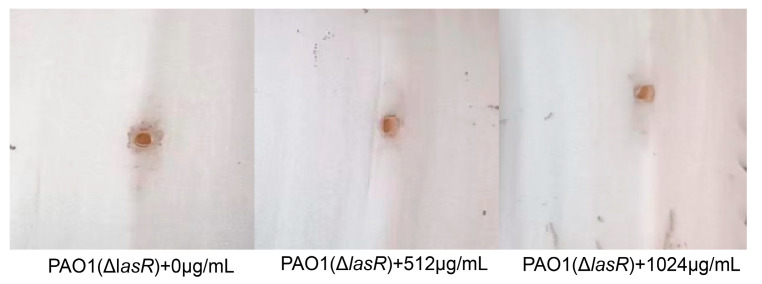
Effect of different concentrations of VA on the pathogenicity of PAO1 (Δ*lasR*).

**Figure 10 microorganisms-12-00985-f010:**
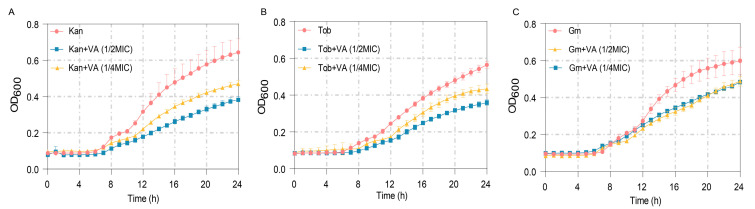
The growth curves of *P. aeruginosa* treated with sub-MIC antibiotics combined with VA at different concentrations. (**A**) Kanamycin (30 µg/mL); (**B**) Tobramycin (1 µg/mL); (**C**) Gentamycin (15 µg/mL).

**Table 1 microorganisms-12-00985-t001:** The strains and plasmids used in the study.

Strains or Plasmids	Description	Reference
Strains		
*P. aeruginosa* (PAO1)	Wild-type strain	This study
*E. coli* OP50	*E. coli* uracil-auxotrophic strain	This study
PAO1 (Δ*lasR*)	*LasR* knocked mutant of PAO1	[27]
PAO1 (Δ*lasI*Δ*rhlI*Δ*pqsA*Δ*rhlR*Δ*pqsR*)	PAO1 quintuple mutant with *lasI*, *rhlI*, *pqsA*, *rhlR* and *pqsR* knocked out	[27]
PAO1 (Δ*lasI*Δ*rhlI*Δ*pqsA* Δ*lasR*Δ*pqsR*)	PAO1 quintuple mutant with *lasI*, *rhlI*, *pqsA*, *lasR* and *pqsR* knocked out	[27]
PAO1 (Δ*lasI*Δ*rhlI*Δ*pqsA* Δ*lasR*Δ*rhlR*)	PAO1 quintuple mutant with *lasI*, *rhlI*, *pqsA*, *lasR* and *rhlR* knocked out	[27]
Plasmids		
PMS402	Expression reporter vector carrying the promoterless luxCDABE; Kan^r^, Tmp^r^	[28]
pKD-*lasI*	pMS402 containing *lasI* promoter region; Kan^r^, Tmp^r^	[29]
pKD-*lasR*	pMS402 containing *lasR* promoter region; Kan^r^, Tmp^r^	[29]
pKD-*rhlI*	pMS402 containing *rhlI* promoter region; Kan^r^, Tmp^r^	[29]
pKD-*rhlR*	pMS402 containing *rhlR* promoter region; Kan^r^, Tmp^r^	[29]
pKD-*pqsA*	pMS402 containing *pqsA* promoter region; Kan^r^, Tmp^r^	[30]
pKD-*pqsR*	pMS402 containing *pqsR* promoter region; Kan^r^, Tmp^r^	[30]
pKD-*lasB*	pMS402 containing *lasB* promoter region; Kan^r^, Tmp^r^	[30]
pKD-*rhlA*	pMS402 containing *rhlA* promoter region; Kan^r^, Tmp^r^	[30]
pKD-*phzA*1	pMS402 containing *phzA1*promoter region; Kan^r^, Tmp^r^	[30]

## Data Availability

Data are contained within the article.

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
