# Peer review of "Veratryl Alcohol Attenuates the Virulence and Pathogenicity of Pseudomonas aeruginosa Mainly via Targeting las Quorum-Sensing System"

_microorganisms, 2024, doi:10.3390/microorganisms12050985_

Round 1
Reviewer 1 Report
Comments and Suggestions for Authors
See attached file.

None.
Author Response
This material is original research and has not been published previously or published elsewhere. The authors have no conflicts of interest to declare. We appreciate your consideration of our manuscript and look forward to receiving the reviewers' comments. If you have any questions, please feel free to contact me.

Reviewer 2 Report
Comments and Suggestions for Authors
Dear authors, thank you for the opportunity to read your work and congratulations on the conceptualization and development of the project.
The topic is extremely relevant and focuses on the investigation of an antimicrobial agent for Pseudomonas aeruginosa, a pathogen classified by the World Health Organization as a priority in the development of new drugs (https://www.who.int/news/item/27-02-2017-who-publishes-list-of-bacteria-for-which-new-antibiotics-are-urgently-needed)
The paper is well written and presents a good discussion, but it is necessary to add control groups of the drug tested (veratryl alcohol) in the tests performed with the invertebrate models and Chinese chard. The methodologies employed were intelligent and enlightening, but the drug can only be chosen as a possible antimicrobial agent if it presents minimal toxicological data.
The addition of control groups will enrich the literature as there is little toxicological data on this alcohol. Only a few studies have reported the use of the drug (Sadaqat B, Khatoon N, Malik AY, Jamal A, Farooq U, Ali MI, He H, Liu FJ, Guo H, Urynowicz M, Wang Q, Huang Z. Enzymatic decolorization of melanin by lignin peroxidase from Phanerochaete chrysosporium. Sci Rep. 2020 Nov 19;10(1):20240. doi: 10.1038/s41598-020-76376-9. PMID: 33214596; PMCID: PMC7677534).
Another questionable point is the biofilm formation time, as only 24 hours of incubation limits us to thinking of planktonic cultures, as well as the Minimum Inhibitory Concentration test. I suggest performing the tests with 48h or more for better signaling and structuring of the P. aeruginosa biofilm. (Pericolini E, Colombari B, Ferretti G, Iseppi R, Ardizzoni A, Girardis M, Sala A, Peppoloni S, Blasi E. Real-time monitoring of Pseudomonas aeruginosa biofilm formation on endotracheal tubes in vitro. BMC Microbiol. 2018 Aug 14;18(1):84. doi: 10.1186/s12866-018-1224-6. PMID: 30107778; PMCID: PMC6092828.)
Please add the control groups, testing only the drug on the survival curve of the larvae and flies.
If possible, test the drug on the 48-hour biofilm. As stated above, time is a major influence on the structuring of the polysaccharide matrix of the P. aeruginosa biofilm. This data will provide the necessary robustness for the drug's novelty.
Author Response
Response to Reviewer 2 Comments:
Thank you very much for your nice patience and valuable comments for our manuscript “Veratryl alcohol attenuates the virulence and pathogenicity of Pseudomonas aeruginosa mainly via targeting las Quorum-Sensing System”. We have carefully revised the manuscript according to the reviewers’ suggestion and all the modification were marked in yellow color in the revised manuscript. The point to point responses are following in details.
Comments 1: Please add the control groups, testing only the drug on the survival curve of the larvae and flies.
Response 1: Thanks for your valuable comments. Actually, we have tested the drug only on the survival curve of the C. elegans and Drosophila melanogaster and the results were shown in the group of PAO1(-)+VA(1024μg/mL) in Figure6 (B) (line 333) and the group of PAO1(-)+VA(1024μg/mL) in Figure6 (C) (line 333) respectively. The results indicated that VA showed little toxicity for C. elegans and Drosophila melanogaster compared with the blank control group of PAO1(-). Also, in the discussion section, we discussed the reason for those shown in lines 428-432.
Comments 2: If possible, test the drug on the 48-hour biofilm. As stated above, time is a major influence on the structuring of the polysaccharide matrix of the P. aeruginosa biofilm. This data will provide the necessary robustness for the drug's novelty.
Response 2: Thanks for your nice suggestion. We have performed the experiment as suggested and the results are added in line 311.

Reviewer 3 Report
Comments and Suggestions for Authors
Reviewer comments:
The reviewed manuscript is entitled: “Veratryl alcohol attenuates the virulence and pathogenicity of Pseudomonas aeruginosa mainly via targeting las Quorum Sensing System”.
The submitted manuscript is well-written. My comments are as follows:
Specific comments.
1. In line 76 (Introduction), there is a grammatical error,…the bacterial growth of bacteria and improve.. Address this and other grammatical errors in the manuscript.
2. If known, the authors need to explain the precise mechanism of action by which Veratryl alcohol attenuates the virulence of P. aeruginosa
3. The authors need to motivate the use of Chinese cabbage, Drosophila melanogaster and C. elegans as infection models for assessing the activity of Veratryl alcohol
4. Under section 2.11 (Statistical analysis), mention the name of the statistical test used to determine statistical significance on GraphPad Prism version 5.0.1 software.
5. The molecular docking results (Figure 7). The diagram and Figure 7 must be on the same page.
6. In line 395 (Discussion), there is a formatting error “A spergillus terreus” should be written as Aspergillus terreus. Correct this and other formatting errors in the manuscript.
7. The limitations of the study need to be included under the Discussion section.
Comments on the Quality of English Language
Moderate editing of the English language is required.
Author Response
Response to Reviewer 3 Comments:
Thank you very much for your nice patience and valuable comments for our manuscript “Veratryl alcohol attenuates the virulence and pathogenicity of Pseudomonas aeruginosa mainly via targeting las Quorum-Sensing System”. We have carefully revised the manuscript according to the reviewers’ suggestion and all the modification were marked in yellow color in the revised manuscript. The point to point responses are following in details.
Comments 1: In line 76 (Introduction), there is a grammatical error,…the bacterial growth of bacteria and improve. Address this and other grammatical errors in the manuscript.
Response 1: Thanks very much for your kind suggestion. We have carefully checked the problem mentioned and correct the errors in the revised manuscript, such as indicated in line 75, line 51 and line 117.
Comments 2: If known, the authors need to explain the precise mechanism of action by which Veratryl alcohol attenuates the virulence of P. aeruginosa.
Response 2: Thank you very much for your valuable suggestion. We have tried to explained the main mechanism of action by which VA attenuates the pathogenesis of PA more precisely in the revised version.
Actually, the docking analysis was used to predict the potential target of VA, LasR protein, on QS inhibition firstly which was shown in section 3.5. Then the gene expression in the QS quintuple mutants was performed to further find out that las system might be the target of VA in section 3.6. Finally,the virulence analysis via Chinese cabbage infection model infected by lasR deletion mutant of PAO1(ΔlasR) confirmed the main target of VA, LasR protein, on QSI In section 3.6.
Comments 3: The authors need to motivate the use of Chinese cabbage, Drosophila melanogaster and C. elegans as infection models for assessing the activity of Veratryl alcohol.
Response 3: Thank you for your nice suggestion. Drosophila melanogaster and C. elegans models have been conventionally used to study the infection and pathogenicity P. aeruginosa in vivo in many previous reports, like “Discovery of psoralen as a quorum sensing inhibitor suppresses Pseudomonas aeruginosa virulence” (DOI:10.1007/s00253-024-13067-9), “Paraoxonase 1, quorum sensing, and P. aeruginosa infection: a novel model’’ (DOI:10.1007/978-1-60761-350-3_17), and “Inhibition of biofilm formation, quorum sensing and infection in Pseudomonas aeruginosa by natural products-inspired organosulfur compounds” (DOI:10.1371/journal.pone.0038492). Chinese cabbage model has also been used to primarily determine the virulence of PA according to the reports “Modeling Pseudomonas aeruginosa pathogenesis in plant hosts (DOI:10.1038/nprot.2008.224)”, “Use of plant and insect hosts to model bacterial pathogenesis(DOI:0.1016/s0076-6879(02)58077-0)”. Our previous published article “Sennoside A inhibits quorum sensing system to attenuate its regulated virulence and pathogenicity via targeting LasR in Pseudomonas aeruginosa” (DOI: 10.3389/fmicb.2022.1042214) applied the Chinese cabbage, Drosophila melanogaster and C. elegans models to assess the activity of Sennoside A. We have mentioned the appropriate references in section 2.7.1, 2.7.2 and 2.7.3 in the revised version.
Comments 4: Under section 2.11 (Statistical analysis), mention the name of the statistical test used to determine statistical significance on GraphPad Prism version 5.0.1 software.
Response 4: Thanks very much for your valuable suggestion. We have added the specific name of the statistical test used on section 2.11.
Comments 5: The molecular docking results (Figure 7). The diagram and Figure 7 must be on the same page.
Response 5: Thanks very much for your kind suggestion. We have put the diagram and Figure 7 in the same page.
Comments 6: In line 395 (Discussion), there is a formatting error “A spergillus terreus” should be written as Aspergillus terreus. Correct this and other formatting errors in the manuscript in the revised version.
Response 6: Thank you very much for your kind suggestion. We have carefully checked the problem and corrected the kind of error in line 409 in the revised version.
Comments 7: The limitations of the study need to be included under the Discussion section.
Response 7: Thanks for your valuable suggestion. The relevant discussion has been added to the discussion section shown in lines 432-435.

Reviewer 4 Report
Comments and Suggestions for Authors
The author presented an interesting works on inhibition of quorum sensing using VA. 1) In the growth experiment (line 240) I would recommend having a known identified QS inhibitor like 'methy gallate', reference: Scientific Reports volume 13, Article number: 17942 (2023) or 'Itaconimide', reference: 'Front Cell Infect Microbiol. 2018; 8: 443' as a positive control.
2) In the effect of VA on elastase experiment, the OD495 plot shows difference between the 512ug/ml and the 1024ug/ml. The author said that the experiment were conducted in triplicates and the figure 3A isnt convincing for the difference observed. Images of triplicate experiments would be convincing.
Author Response
Response to Reviewer 4 Comments:
Thank you very much for your nice patience and valuable comments for our manuscript “Veratryl alcohol attenuates the virulence and pathogenicity of Pseudomonas aeruginosa mainly via targeting las Quorum-Sensing System”. We have carefully revised the manuscript according to the reviewers’ suggestion and all the modification were marked in yellow color in the revised manuscript. The point to point responses are following in details.
Comments 1: The author presented an interesting works on inhibition of quorum sensing using VA. 1) In the growth experiment (line 240) I would recommend having a known identified QS inhibitor like 'methy gallate', reference: Scientific Reports volume 13, Article number: 17942 (2023) or 'Itaconimide', reference: 'Front Cell Infect Microbiol. 2018; 8: 443' as a positive control.
Response 1: Thank you very much for your kind suggestion. Receiving your valuable suggestion, we immediately contacted the reagent vendor to purchase the mentioned positive control drugs. However, the reagents would take three days more to be obtained and the supplemental test would take two days. It is a bit of difficult for us to supplement the experiment before the deadline.
The aim of growth experiment (line 263) was to obtain the appropriate concentrations of VA in which VA did not inhibit the growth of PA and then guarantee the QSI effect of VA on PA was derived from the inhibition of VA on QS system only but not from the inhibited growth of PA. Many reports on QSIs contained no other positive control for subinhibitory concentration determination, like the articles “Falcarindiol Isolated from Notopterygium incisum Inhibits the Quorum Sensing of Pseudomonas aeruginosa” (DOI: 10.3390/molecules26195896), “Baicalin inhibits biofilm formation, attenuates the quorum sensing-controlled virulence and enhances Pseudomonas aeruginosa clearance in a mouse peritoneal implant infection model” (DOI: 10.1371/journal.pone.0176883) and “Paeonol Attenuates Quorum-Sensing Regulated Virulence and Biofilm Formation in Pseudomonas aeruginosa” (DOI: 10.3389/fmicb.2021.692474).
Anyway, the suggestion pushes us to consider adding some positive drug controls in pathogenesis analysis to improve the further research work and our publication. Thanks again for your valuable suggestion.
Comments 2: In the effect of VA on elastase experiment, the OD495 plot shows difference between the 512ug/ml and the 1024ug/ml. The author said that the experiment were conducted in triplicates and the figure 3A isn’t convincing for the difference observed. Images of triplicate experiments would be convincing.
Response 2: Thank you very much for your suggestion. Two additional images were showed below and could be supplemented in the revised version if necessary.
Figure3. Effects of VA on elastase production in P. aeruginosa.

Round 2
Reviewer 1 Report
Comments and Suggestions for Authors
Thanks to the authors, they attended all comments and improved the paper.
Reviewer 2 Report
Comments and Suggestions for Authors
Thank you for the opportunity to view this work. All the notes I made were completely answered and the results added were very enlightening.
kind regards.